# Age- and Gender-Related Femoral Bowing Analysis in the Korean Population and Features for Clinical Applications

**DOI:** 10.3390/medicina60121930

**Published:** 2024-11-23

**Authors:** Ju-Yeong Kim, Gyu-Min Kong

**Affiliations:** 1Department of Orthopedic Surgery, Gyeongsang National University Changwon Hospital, Gyeongsang National University School of Medicine, 11 Samjeongja-ro, Seongsan-gu, Changwon 51472, Republic of Korea; passion0309@hanmail.net; 2Department of Orthopedic Surgery, Inje University Haeundae Paik Hospital, 875 Haeun-daero, Haeundae-gu, Busan 48108, Republic of Korea

**Keywords:** femur, bowing, Korean population, atypical fracture

## Abstract

*Background and Objectives*: The anterolateral bowing of the femur shows differences between races and has recently caused many clinical problems. Asians tend to have increased femoral bowing, but there is a lack of large-scale studies. We aim to identify the patterns of femoral bowing in the Korean population through comprehensive analysis and address its clinical implications. *Materials and Methods*: We analyzed 550 femoral radiographs from Korean patients using three different views: anteroposterior, lateral, and 15-degree internal rotation. Initial univariate analysis examined age and gender differences, followed by multivariate analysis incorporating height and weight to understand their combined effects on femoral bowing. *Results*: The study included 229 (41.6%) males and 321 (58.4%) females, with a mean age of 62.53 years (SD = 21.93). Initial analysis showed greater femoral bowing in females than males by 2.72° (*p* < 0.001) in anteroposterior views. However, multivariate analysis revealed age to be the primary significant factor affecting femoral bowing across all viewing angles (*p* < 0.001), while gender effects became non-significant when controlling for other variables. The AP angle regression model explained 26% of the total variance, with each year increase in age associated with a 0.12-degree increase in bowing angle. *Conclusions*: This study demonstrated that age is the primary factor influencing femoral bowing in the Korean population, with apparent gender differences potentially attributable to age distribution differences between groups. Anteroposterior radiographic imaging proved most suitable for assessing bowing angles. These findings provide important insights for surgical planning and implant selection, particularly in addressing potential mismatch issues in Asian populations.

## 1. Introduction

While the etiology of atypical femoral fractures (AFFs) has largely been attributed to bisphosphonate-related osteoporosis medications, recent studies have highlighted the importance of multifactorial causes, including changes in anterolateral femoral bowing [1]. The unique characteristics of femoral bowing in Asian populations have been identified as potential risk factors for AFF [2]. This distinct geometry of the lower limbs has emerged as a risk factor for AFF, as abnormal limb geometry is thought to increase stress on the lateral cortex of the femoral shaft, thereby heightening mechanical fatigue and contributing to the development of AFF. Several studies focused on Asian populations support this association [3]. Notably, statistical data indicate that the risk of AFF in Asian populations is five times higher compared to that in Caucasians [4].

Femoral bowing is not only a risk factor for atypical fractures but can also impact the outcomes of various surgical procedures such as intramedullary nailing and arthroplasty [5]. In clinical practice, the primary concern is often the mismatch between orthopedic implants and increased anterolateral femoral bowing, rather than the precise etiology [5,6,7,8]. Severe deformities of femoral bowing can affect limb alignment and implant fit, potentially leading to iatrogenic fractures [9]. Abnormal femoral bowing has been identified as a risk factor in various conditions [10]. Since AFF are known to be a type of insufficiency fracture, it is essential to cover the entire femoral shaft when using implants for fracture fixation to mitigate stress concentration effects on the lateral cortex at the implant’s endpoint, as might occur with a short plate. For these reasons, intramedullary nailing is preferred in the treatment of AFF [11]. Unlike extramedullary implants, intramedullary nailing is significantly influenced by the geometry of the femur, especially the shape of the medullary canal. Consequently, structural mismatches between intramedullary nails and the geometry of the femur are associated with a higher incidence of surgical complications, often requiring revision surgeries [12].

The significance of varus limb alignment associated with osteoarthritis, which is increasingly prominent in East Asia, has recently gained attention. This association strongly suggests a link between osteoarthritis and femoral bowing. Studies have reported that patients with advanced varus knee osteoarthritis exhibit significant progression in femoral bowing [13]. Changes in limb alignment due to various pathological or physiological causes have been found to strongly correlate with the onset and progression of knee osteoarthritis [14]. For osteoarthritis patients, the survival rate of total knee arthroplasty (TKA) largely depends on the correct restoration of the limb’s mechanical axis and the accuracy of bone resection. Femoral and tibial cuts should ideally be perpendicular to their respective mechanical axes in the coronal plane, with a margin of error within three degrees from the ideal position. The importance of restoring mechanical axis alignment postoperatively in TKA has long been emphasized [15]. In patients with severe femoral bowing, performing TKA may present issues of implant mismatch when using standard instruments. This is clinically significant as nearly all commercial TKA systems are based on Western osteometric models [16].

For these reasons, recent studies have investigated trends in femoral bowing changes across different ethnicities and nationalities, reporting variations in bowing [17,18,19]. Comparisons between Asian and Western populations have revealed ethnicity-specific characteristics in femoral bowing [14,16]. Comparative studies of different racial groups in the United States have found that Asian populations exhibit more pronounced femoral bowing and a higher risk of AFF compared to other ethnicities [4]. Asian women tend to have more curved femurs than Caucasians, while African populations generally show the straightest femoral alignment, which may partly explain the higher prevalence of AFF in Asia [20]. However, large-scale studies on femoral bowing angles in the Korean population using plain radiographs are lacking. Predicting femoral bowing through plain radiography, the most commonly used imaging modality in clinical practice, could offer an economical and practical tool to aid clinical decision-making. Previous studies have reported that Asian populations exhibit greater femoral bowing angles compared to Western populations, which significantly influences implant design and surgical strategies. This study aims to validate these characteristics in the Korean population, thereby contributing to the existing knowledge on femoral bowing in Asian cohorts.

With the global trend of population aging, South Korea is becoming an ultra-aged society at an unprecedented rate, making cardiovascular and musculoskeletal diseases significant socioeconomic issues. In the musculoskeletal field, not only degenerative hip and knee varus deformities but also increased anterolateral bowing of the femoral shaft have led to the continued occurrence of AFF. Numerous studies on AFF [1,5,7,21] have revealed that increased anterolateral bowing is observed clinically, even among elderly individuals and middle-aged women who have not experienced fractures.

Therefore, this study aims to conduct a large-scale retrospective analysis of the Korean population to measure anterolateral femoral bowing in adults without AFF and to analyze the degree of change according to age and gender. Using plain radiography, the most commonly performed imaging method in clinical practice, to predict femoral bowing may help to assess the risk of AFF and assist in preoperative planning. Through this approach, we aim not only to establish a comprehensive database for the Korean population but also to extract clinically relevant insights, including optimized surgical treatment strategies, by analyzing and comparing anterolateral bowing in patients with AFF.

## 2. Materials and Methods

### 2.1. Study Design

This cross-sectional study was designed to investigate femoral bowing in the Korean population. Subjects were recruited from patients who underwent femoral radiographic examinations at Gyeongsang National University Changwon Hospital between 1 March 2016 and 31 December 2022. Participants were grouped according to gender and age. The degree of femoral bowing was measured on anteroposterior (AP), lateral, and 15-degree internal rotation views. Subsequently, the collected data were statistically analyzed and visualized to examine changes in femoral bowing according to age and gender, determine which examination method is more significant, and identify trends in femoral bowing.

### 2.2. Participant Selection

To analyze femoral bowing using plain radiographs, we included radiographic images from patients who had undergone full-length femoral imaging specifically of the femur in AP, lateral, and 15-degree internal rotation views during their orthopedic evaluation. In our orthopedic department, radiographic examinations are commonly performed to assess various complaints, including pain or discomfort, even when no specific pathology is initially identified. During the specified period, 606 individuals underwent radiographic examinations of the femur, and after excluding images with missing views or poor quality, a final set of 550 single images from 550 different patients was included in the analysis. No restrictions were placed on age, gender, height, or weight. Exclusion criteria included previous surgery on the femur or adjacent joints, pathological conditions such as fractures, bone tumors, congenital malformations, and non-Korean nationality. Additionally, the radiographic images were evaluated for the appropriate quality necessary for measurements. After applying these criteria, 56 images were excluded due to poor image quality or missing views, resulting in a final analysis set of 550 radiographic examinations from 550 different individuals (229 males and 321 females) of Korean nationality. All imaging was performed at a single institution, Gyeongsang National University Changwon Hospital, during the specified period.

### 2.3. Bowing Measurement

#### 2.3.1. Definition of Femoral Shaft

Using the AP femoral images, the femoral shaft was defined as the length from just below the lesser trochanter to just above the femoral condyles and divided into three equal parts (proximal, middle, and distal).

#### 2.3.2. Standardized Patient Positioning

For the 15-degree internal rotation view, which is a standard radiographic technique used in orthopedic assessment, patients were positioned supine on the X-ray table with their affected leg internally rotated to 15 degrees from the neutral position. This standardized position allowed for consistent and comparable measurements across all patients while providing optimal visualization of the femoral shaft curvature.

#### 2.3.3. Measurement Technique

To measure the anterolateral bowing angle, 50 mm long lines bisecting the medullary canal (proximal and distal bisecting lines) were placed at the proximal and distal ends of the defined shaft. The angle formed by the intersection of these two lines was used for measurement. A positive value was defined for bowing in the anterolateral direction, and a negative value was defined for posteromedial bowing. In cases where one leg had previous surgery or pathology, measurements were taken from the healthy contralateral side. All images were captured and measured based on full-length views when possible. This method was applied to AP, lateral, and 15-degree internal rotation views, and the correlation between these results was analyzed (Figure 1).

#### 2.3.4. Quality Control

All measurements were manually performed by a single experienced clinician over a one-month period based on commonly used techniques described in previous studies [22]. To ensure measurement reliability, a second clinician independently measured a random sample of 50 cases, and inter-observer reliability was assessed.

### 2.4. Variables

The degree of femoral bowing was measured from AP, lateral, and 15-degree internal rotation views. Age, gender, height, and weight were recorded, and we sought to identify trends in these variables and correlations between them. Age has been shown to correlate with femoral bowing in several previous studies. In this study, we also examined the correlation between age and femoral bowing for each imaging view in our sample, determining its significance in the Korean population. Participants were grouped by gender to investigate differences in femoral bowing between males and females.

### 2.5. Statistical Analysis

Statistical analyses were performed using IBM SPSS Statistics software (version 24.0 for Windows; IBM Corp., Armonk, NY, USA) and Python with statsmodels library for extended analyses. In anatomical variation studies, systematic statistical analysis is essential for validating findings and ensuring their clinical application. Our analysis proceeded in two complementary phases: First, we examined basic measurements and established patterns of femoral bowing variation by age and gender through direct comparisons and correlation analyses. Then, we advanced to a comprehensive multivariate analysis to understand how multiple anthropometric factors simultaneously influence femoral morphology. This systematic approach allowed for both the clear demonstration of fundamental relationships and a deeper understanding of complex interactions between variables.

For the primary analysis, data normality was first assessed to determine appropriate analytical methods. Through univariate analysis, the basic relationships between individual variables were examined to identify fundamental trends within the dataset. Variance analysis of each variable with femoral bowing was conducted, with *p*-values less than 0.05 considered statistically significant. Pearson correlation coefficients were used to determine the degree of correlation between age and femoral bowing angle, interpreted as weak (<0.3), moderate (0.3–0.5), or strong (>0.5) to provide a meaningful interpretation of relationship strengths beyond statistical significance. Independent *t*-tests were used to determine significant differences in femoral bowing angles between genders, with two-sided *p*-values less than 0.05 considered significant.

For the extended analysis, multiple regression models were constructed to examine the combined effects of age, gender, height, and weight on femoral bowing angles. This approach allowed for comprehensive understanding of how multiple factors simultaneously influence femoral morphology. Three separate models were developed for AP, IR, and LAT angles. The regression models included gender (coded as 0 for female, 1 for male), age (in years), height (in centimeters), and weight (in kilograms) as independent variables. This comprehensive analytical strategy not only enhanced the robustness of the study findings but also facilitated the elucidation of complex interactions among the variables.

Model results were visualized using scatter plots with regression lines and confidence intervals for continuous variables. These analytical steps were essential for transforming complex biomedical data into clinically practical insights and aided in intuitively understanding the relationships.

## 3. Results

### 3.1. Patient Demographics and Femoral Bowing Measurements

From March 2016 to December 2022, a total of 606 radiographic examinations were performed. After applying exclusion criteria and quality assessment, 550 radiographs were included in the final analysis. The baseline characteristics and measurements of the study population are summarized in Table 1. The sample comprised 229 (41.6%) males and 321 (58.4%) females, with a mean age of 62.53 years (SD = 21.93). Age distribution showed considerable variation, ranging from 3 to 95 years, with females generally being older (mean age 69.50 years) than males (mean age 53.07 years).

Femoral bowing measurements varied across the three radiographic views. In the anteroposterior view, the bowing angle ranged from −11 to +24 degrees (mean = 2.93°, SD = 5.51°), with notable differences between males (mean = 1.11°, SD = 4.21°) and females (mean = 4.24°, SD = 5.96°). The 15-degree internal rotation view showed angles ranging from −16 to +21 degrees (mean = −4.12°, SD = 5.77°), while the lateral view measurements ranged from +3 to +29 degrees (mean = 14.14°, SD = 3.66°). These measurements demonstrated consistent patterns across different viewing angles, with females generally showing greater bowing angles than males across all views (Table 1).

### 3.2. Age- and Gender-Related Trends in Femoral Bowing

The degree of femoral bowing, as measured via anteroposterior radiographs, demonstrates a clear trend across age groups and genders. Analysis reveals that bowing tends to increase with age, with a more pronounced increase observed in females compared to males. This pattern is consistent across all age brackets, with older groups showing significantly higher degrees of bowing than younger cohorts (Table 2).

### 3.3. Correlation Analysis Between Age and Femoral Bowing

Pearson correlation coefficients showed a positive relationship between age and femoral bowing angle in the sample. The coefficients differed for anteroposterior, internal rotation, and lateral views. The anteroposterior and internal rotation views showed moderate positive correlations (r = 0.483 and r = 0.462, respectively, both *p* < 0.001). Although these correlations were statistically significant, they represented moderate-strength relationships, explaining approximately 23% and 21% of the variance in bowing angles, respectively. The lateral view demonstrated a significant but weak positive correlation (r = 0.206, *p* < 0.001), accounting for only about 4% of the variance (Table 3). These results suggest that anteroposterior or internal rotation images have stronger, though still moderate, correlations compared to lateral radiographs (Figure 2).

### 3.4. Gender Differences Analysis

When compared using an independent *t*-test, the femoral bowing angle was, on average, 2.72° greater in females than in males (*p*-value < 0.001). There was also a statistically significant age difference between males and females in the sample (Table 4).

### 3.5. Multiple Regression Analysis

Regarding gender, B = 1.374 (*p*-value <0.05), rejecting the null hypothesis and accepting the alternative hypothesis, indicating that gender significantly affects changes in femoral bowing. The positive β sign indicates that being female increases the degree of femoral bowing by 1.374. Age group is also statistically significant, with B = 0.108 (*p*-value < 0.001). For each increase in age group, the degree of femoral bowing increases by 0.108. To determine the relative influence of gender and age group on femoral bowing, we compare the standardized coefficient β values. With gender β = 0.123 and age group β = 0.435, age has a relatively higher influence on the degree of femoral bowing than gender (Table 5).

### 3.6. Extended Multivariate Analysis Results

To comprehensively understand the relationships between femoral bowing and multiple variables, we conducted an extended multivariate analysis. From the initial 550 patients analyzed in this study, after excluding cases with missing height or weight data, 457 patients (83.1%) were included from this multivariate analysis. Multiple regression analysis was performed separately for AP, IR, and LAT angles (Table 6). A multivariate analysis was performed to examine the effects of age, gender, height, and weight on femoral bowing angles. Confidence intervals for the mean values and comparative significance levels were calculated using *p*-values, enhancing the rigor of this study.

Here, we discuss the coefficients (β) and *p*-values for each variable’s effect on femoral bowing angles. Gender was coded as 0 (female) and 1 (male). Significant *p*-values (*p* < 0.05) are shown in bold. R^2^ indicates the proportion of variance explained by the model for each angle measurement.

The AP angle regression model explained 26% of the total variance. Age demonstrated the strongest influence, with each year increase associated with a 0.124-degree increase in bowing angle (β = 0.12, *p* < 0.001). Gender, however, did not show a statistically significant effect (β = −0.82, *p* = 0.153), and height showed a slight negative correlation (β = −0.055, *p* = 0.028). Weight had minimal influence (β = 0.00, *p* = 0.971) and was not statistically significant.

For IR angle measurements, the model explained 23% of the variance, with age once again showing a strong influence. Each year increase in age was associated with a 0.116-degree increase in the IR angle (β = 0.12, *p* < 0.001). Height maintained a slight negative correlation (β = −0.06, *p* = 0.034), while neither gender (β = −1.06, *p* = 0.083) nor weight (β = 0.00, *p* = 0.838) showed significant effects.

The LAT angle model accounted for the lowest explanatory power, explaining only 6% of the variance. Age remained a significant predictor, with each year increase associated with a 0.036-degree increase in LAT angle (β = 0.04, *p* < 0.001). Gender was not statistically significant (β = −0.63, *p* = 0.148), and neither height (β = −0.03, *p* = 0.189) nor weight (β = 0.01, *p* = 0.399) showed meaningful influence on the LAT angle.

The analysis indicated a consistent pattern where age showed a significant linear relationship with all femoral bowing measurements (Figure 3). While gender was included as a variable in the regression models, its influence was not statistically significant across AP, IR, and LAT angles. Overall, these results suggested that age was the primary predictor of femoral bowing across AP, IR, and LAT angles, while height showed a minor but consistent negative association, and weight had no substantial impact.

## 4. Discussion

Through the analysis of 550 full-length femoral radiographs from a Korean population sample, we confirmed the presence of femoral bowing (mean = 2.93°, SD = 5.51°). Initial univariate analysis showed that femoral bowing angles were greater in females than in males by 2.72° (*p*-value < 0.001). Our subsequent multivariate analysis, which adjusted for height, weight, and age, revealed that age was the primary significant factor affecting femoral bowing across all viewing angles, while gender effects became non-significant when controlling for other variables. This finding suggests that the apparent gender differences might be influenced by age distribution differences between male and female groups in our sample, particularly in those over 50 years old. In the group with the most severe bowing, the maximum curvature was measured at 8.52°. This age-related pattern in femoral bowing, initially appearing as gender differences, may be associated with osteoporotic degeneration occurring in postmenopausal women, as other studies have reported that elderly Asian women exhibit more severe femoral bowing compared to younger women or elderly men [23]. While there may be potential associations with bisphosphonate (BP) use, a history of osteoporosis, or hormonal influences, current studies primarily interpret this as an ethnic characteristic. Differences in femoral bowing have been observed between Western and Asian populations, with particularly pronounced bowing in older female groups within Asian populations. This is most reliably explained by genetic factors and ethnic characteristics. Anteroposterior and 15-degree internal rotation radiographs showed more significant correlations with femoral bowing. However, considering imaging consistency, measurements obtained from anteroposterior imaging are deemed more reliable. Clinical application was limited in cases where the degree of bowing was underestimated due to variations in internal rotation during imaging. Additionally, we developed a predictive model incorporating age, gender, height, and weight, which demonstrated that age is the most reliable predictor of femoral bowing changes.

An abnormal increase in femoral bowing poses several clinical risks. The mismatch between orthopedic implants and the femur caused by excessive bowing can lead to technical problems and complications such as angular deficits, iatrogenic fractures, and instrument penetration [24]. Most patients with AFF exhibit increased anterolateral bowing and femoral shapes that do not align well with intramedullary nails, highlighting the importance of accurate preoperative assessment [25]. Excessive femoral bowing during intramedullary nailing for fracture fixation can lead to anterior cortical penetration during nail insertion and challenges in achieving proper fracture reduction. The findings of this study suggest the necessity of designing customized intramedullary nails with larger radii of curvature tailored for Asian populations. Such modifications could potentially minimize complications associated with implant malalignment during surgical procedures.

Femoral bowing has a direct impact on TKA in patients with osteoarthritis. To restore limb alignment, the use of an intramedullary guide for distal femoral resection is the standard procedure in TKA. Most instrumentation systems do not account for femoral bowing and provide a standard six-degree cutting block to align with the commonly reported physiological valgus angle of six degrees. To create a distal femoral cut perpendicular to the mechanical axis of the femur, the choice of cutting block should be individualized based on preoperative planning using full-length weight-bearing radiographs of the limb. Regional differences in the epidemiology of osteoarthritis are well documented; the knee-to-hip osteoarthritis ratio is 9:1 in Asians, 3:1 in White Americans, and 1:2 in Swedes. Differences in limb alignment among ethnic groups may contribute to these ratio differences. It is hypothesized that the higher knee joint inclination angle in Asian adults may contribute to the higher knee-to-hip osteoarthritis ratio observed in Asian populations, though further studies are required to confirm this [26]. The osteometry of Asians differs from that of Western populations in both size and shape. This is clinically significant, as nearly all commercial TKA systems are based on osteometric models of Western populations [16]. In TKA, increased femoral bowing can result in malalignment when using standard six-degree cutting blocks and a heightened risk of notching or cortical perforation during intramedullary guide insertion. These factors may contribute to abnormal stress distribution, potentially leading to early implant failure. Therefore, TKA systems should offer additional cutting angle options tailored to the femoral bowing characteristics of the Korean population.

The accurate evaluation of femoral bowing before surgery is crucial not only for total joint arthroplasty but also for fracture surgery [27]. Unlike extramedullary implants, intramedullary nailing is strongly influenced by the geometric structure of the femur, particularly the shape of the medullary canal. Consequently, structural mismatches between intramedullary nails and femoral geometry are associated with a high incidence of surgical complications that may require revision surgeries [12]. Because bone length and bowing vary among ethnicities, implants designed for Western populations may be unsuitable for Asian patients. Through this study, by understanding the degree of bowing in a specific ethnic group, we can propose implant models that better match the femoral morphology of patients from that specific ethnic group. Furthermore, increased femoral bowing itself is a risk factor for patients. Many elderly women exhibit abnormally increased femoral bowing, which is directly linked to the occurrence of AFF. Our study suggests that the greater femoral bowing observed in elderly women may be associated with decreased bone mineral density and hormonal changes following menopause. Differences in pelvic anatomy and muscle mass in females may also contribute as factors influencing femoral bowing. Further research is needed to investigate this association in more detail. Most studies report that changes in load-bearing points due to femoral bowing lead to AFF [28]. As femoral bowing increases, tension acts anterolaterally, concentrating stress in that area and creating conditions conducive to fracture. It has been reported that bowing can affect the characteristics and location of potential fractures [29]. While various factors contribute to AFF, femoral bowing is emphasized as a significant mechanical factor [23]. The geometric characteristics of the femur in Asian populations may explain the higher incidence of diaphyseal AFF in Asians, even among patients who have not used bisphosphonates (BP). Although excessive suppression of bone remodeling due to BP use is still considered a primary cause of AFF, these anatomical characteristics may also play a contributory role [30].

In our study, 7 out of 550 patients experienced AFF, all of whom were over the age of 70 with bowing angles between 6° and 16°. Our multivariate analysis underscored age and femoral bowing as key risk factors for atypical fractures, suggesting that increased bowing angles, particularly in older adults, elevate fracture risk. Although our sample size limited the statistical power to derive a definitive risk threshold, future studies with larger populations may establish a meaningful predictive angle.

Nevertheless, this study has limitations. First, the sample size was limited. Larger datasets are necessary to enhance statistical validity, especially for continuous variables. Our reliance on examinations from a single institution and the selection of samples with complete imaging of the unaffected side constrained data acquisition. Second, there is no standardized method for measuring femoral bowing. Various measurement techniques have been proposed in other studies, but errors can arise due to the femur’s three-dimensional structure, making consistent measurement challenging. In this study, we used angle measurements from radiographic imaging as the standard, emphasizing the importance of accurate imaging and consistent angle measurement. However, issues such as patient rotation during imaging and individual alignment characteristics introduced some degree of error due to the lack of uniform criteria. Third, there is a possibility of selection bias in recruiting patient samples. This study did not employ complete random sampling; instead, radiographic imaging was conducted on patients visiting the outpatient clinic. Consequently, patients with certain comorbidities may be over-represented, which could excessively influence the study results. This sampling method limits the generalizability of the findings and carries the risk of introducing biased results. Future research should focus on developing the most accurate methods for measuring femoral bowing and improving the reliability of radiographic examinations to minimize errors.

## 5. Conclusions

Our research on the Korean population demonstrated the complex relationship between age and gender in femoral bowing changes. While age emerged as the primary factor in multivariate analysis, the interaction between age and gender, particularly in older age groups, suggests potential clinical implications that warrant further investigation. In evaluating these changes, we found anteroposterior radiographic images to be particularly suitable for assessing the degree of bowing. This comprehensive approach can aid in evaluating the risk of diseases such as AFF and osteoarthritis and be highly beneficial for addressing potential implant mismatch issues during surgery.

## Figures and Tables

**Figure 1 medicina-60-01930-f001:**
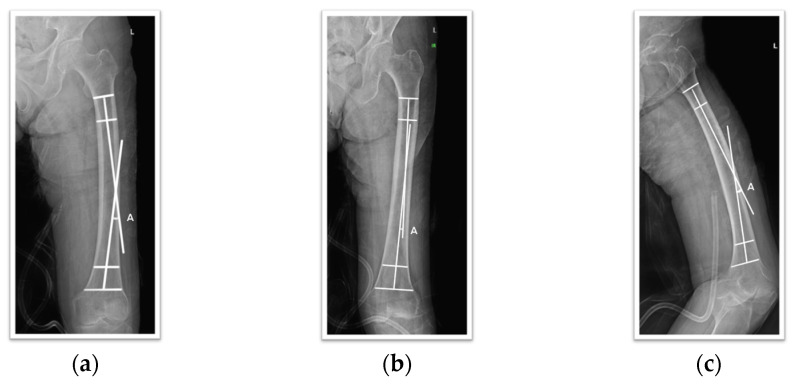
An example of the measurement method for femoral bowing: (**a**) AP view; (**b**) IR view; (**c**) Lat view.

**Figure 2 medicina-60-01930-f002:**
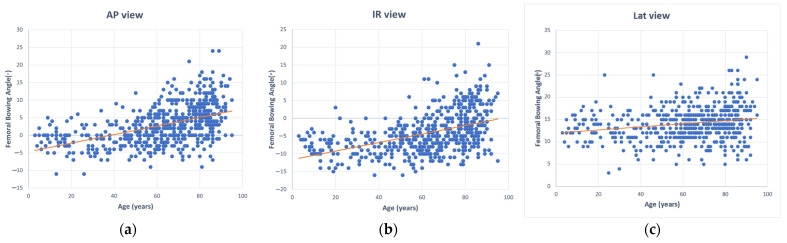
Scatter plots showing the relationship between age and femoral bowing angle: (**a**) age versus femoral bowing angle in anteroposterior (AP) view; (**b**) age versus femoral bowing angle in 15-degree internal rotation (IR) view; (**c**) age versus femoral bowing angle in lateral (Lat) view. the blue dots represent individual data points for each variable, while the red line indicates the regression line representing the correlation.

**Figure 3 medicina-60-01930-f003:**
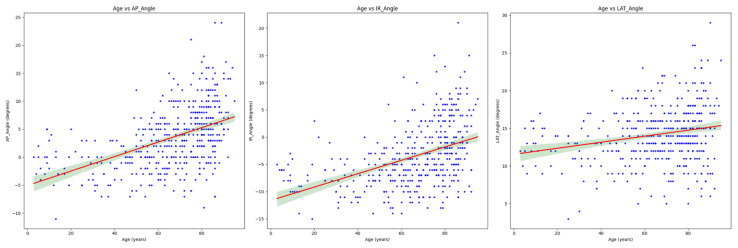
Age-related changes in femoral bowing angles from multivariate analysis. While Figure 2 shows a simple correlation, this figure presents the relationship between age and femoral bowing angles after controlling for gender, height, and weight through multivariate regression analysis. Among all the variables analyzed, only age showed consistently significant associations (*p* < 0.001) across all angle measurements (AP, IR, LAT). The red line represents the fitted regression line adjusting for other variables, and the green shaded area indicates the 95% confidence interval.

**Table 1 medicina-60-01930-t001:** A summary of the characteristics of all patients, males, and females (STROBE).

Variables	All (*n* = 550)	Male (*n* = 229)	Female (*n* = 321)
Age (Years)			
Mean (SD)	62.53 (21.93)	53.07 (22.53)	69.50 (18.65)
Median (IQR)	67 (29)	57 (32)	75 (22)
Range	3~95	3~92	4~95
Angle of femoral bowing (AP, ◦)			
Mean (SD)	2.93 (5.51)	1.11 (4.21)	4.24 (5.96)
Median (IQR)	3 (7)	1 (6)	4 (8)
Range	−11~24	−11 ~ 12	−9~24
Angle of femoral bowing (IR, ◦)			
Mean (SD)	−4.12 (5.77)	−6.16 (4.75)	−2.66 (5.99)
Median (IQR)	−5 (7)	−6 (7)	−4 (8)
Range	−16~21	−16~11	−15~21
Angle of femoral bowing (Lat, ◦)			
Mean (SD)	14.14 (3.66)	13.41 (3.37)	14.67 (3.78)
Median (IQR)	14 (4)	13 (5)	15 (5)
Range	3~29	3~25	5~29

**Table 2 medicina-60-01930-t002:** Correlation between age and femoral AP bowing angle.

Age Group	Total Mean (SD)	Male Mean (SD)	Female Mean (SD)
0–10	−0.60 (2.95)	−0.56 (2.83)	−0.67 (3.39)
10–20	−1.29 (3.22)	−1.32 (3.14)	−1.17 (2.64)
20–30	−1.62 (4.27)	−1.64 (4.99)	−1.57 (2.64)
30–40	−1.60 (3.64)	−1.14 (3.89)	−3.00 (2.52)
40–50	0.00 (3.61)	0.85 (3.75)	−1.47 (2.90)
50–60	0.82 (4.04)	1.43 (3.82)	0.21 (4.22)
60–70	2.98 (4.81)	1.68 (3.62)	4.14 (5.44)
70–80	3.89 (5.03)	2.33 (4.95)	4.60 (4.94)
80–90	6.19 (5.78)	3.81 (3.88)	6.73 (6.01)
90–100	8.52 (5.01)	3.75 (2.50)	9.65 (4.81)

**Table 3 medicina-60-01930-t003:** Correlation between age and femoral bowing angle.

View	Pearson Correlation	*p*-Value
AP	0.483	<0.001
IR	0.462	<0.001
Lat	0.206	<0.001

**Table 4 medicina-60-01930-t004:** Independent *t*-test results for gender differences.

Variable	Mean Difference	T-Score	95% CI	*p*-Value
Age (years)	14.82	6.159	10.066–19.576	<0.001
AP angle (◦)	2.72	4.636	1.559–3.876	<0.001
IR angle (◦)	3.25	4.925	1.942–4.550	<0.001
Lat angle (◦)	1.03	2.441	0.196–1.862	0.016

**Table 5 medicina-60-01930-t005:** Multiple linear regression analysis for femoral bowing (AP) considering age and gender variables.

Variable	Unst.B	SE	Std.β	t-Value	*p*-Value	TOL	VIF
Constant	−4.120	0.558		−7.385	<0.01		
Gender	1.374	0.432	0.123	3.178	<0.1	0.870	1.149
Age Group	0.108	0.010	0.435	11.242	<0.01	0.870	1.149

Unst.B, Unstandardized Beta Coefficient; SE, Standard Error; Std.β, Standardized Beta Coefficient; TOL, Tolerance; VIF, Variance Inflation Factor; F-value: 93.197 (*p*-value <0.01); Adjusted R^2^: 0.240; Durbin–Watson: 2.091.

**Table 6 medicina-60-01930-t006:** Multiple regression analysis of femoral bowing angles with demographic and anthropometric variables.

Variable	AP Angle Coefficient	AP *p*-Value	IR Angle Coefficient	IR *p*-Value	LAT Angle Coefficient	LAT *p*-Value
Intercept	4.20	0.190	−2.22	0.514	15.17	<0.001
Gender	−0.82	0.153	−1.06	0.083	−0.63	0.148
Age	0.12	<0.001	0.12	<0.001	0.04	<0.001
Height (cm)	−0.06	0.028	−0.06	0.034	−0.03	0.189
Weight (kg)	0.00	0.971	0.00	0.838	0.01	0.399
R^2^	0.26		0.23		0.06	

## Data Availability

The data presented in this study are not publicly available due to privacy and ethical restrictions. However, data may be available on reasonable request to the corresponding author, as this study has been exempted from review by the Institutional Review Board (IRB) of Gyeongsang National University Changwon Hospital (GNUCH).

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
