# Peer review of "Age- and Gender-Related Femoral Bowing Analysis in the Korean Population and Features for Clinical Applications"

_medicina, 2024, doi:10.3390/medicina60121930_

Round 1
Reviewer 1 Report
Comments and Suggestions for Authors
Trends in femoral bowing changes across different countries/ continents are important, they can influence the results of surgery, implant fitting , etc.
Interesting paper, however I have some remarks.
Abstract - clear. Introduction - concise, goal of the study clearly stated. Material & methods: angles of bowing (table 1) should be put in the RESULTS section. I do not understand : there were 550 patients or 550 radiographs?
Discussion: among limitation of the study, authors should write that radiographs were taken due to some reasons (trauma? artrhrosis?). This may create a bias.
Author Response
'Trends in femoral bowing changes across different countries/ continents are important, they can influence the results of surgery, implant fitting , etc.
Interesting paper, however I have some remarks.
1.Abstract - clear.
2.Introduction - concise, goal of the study clearly stated.
3.Material & methods: angles of bowing (table 1) should be put in the RESULTS section. I do not understand : there were 550 patients or 550 radiographs?
4.Discussion: among limitation of the study, authors should write that radiographs were taken due to some reasons (trauma? artrhrosis?). This may create a bias.’
Thank you for providing such a valuable comment. Following your suggestions, I have reviewed and made the following modifications to the manuscript:
1.Abstract : I have reviewed the feedback you provided. I have made some adjustments to further improve it.
2.Introduction : I have reviewed the feedback you provided. I have made some adjustments to further improve it.
3.Material & methods: As suggested, we have moved the angle measurements data to the Results section. We have also clarified that our study included 550 single radiographic examinations from 550 different patients. In our orthopedic department, radiographic examinations are commonly performed to assess various complaints, including pain or discomfort, even when no specific pathology is initially identified.
4.Discussion: Thank you for your valuable input. We acknowledge the possibility of selection bias in recruiting patient samples. As we did not employ complete random sampling but conducted radiographic imaging on patients visiting the outpatient clinic, there is a potential for overrepresentation of certain comorbidities or excessive outcome values. We have accordingly specified this as an additional limitation.
Again, I appreciate your insightful feedback, and I have highlighted the changes made in Blue text in the manuscript.
Reviewer 2 Report
Comments and Suggestions for Authors
1. The study's implications on implant mismatch are acknowledged, but more detailed discussion on how the findings could directly inform surgical practices, implant selection, or design adjustments would improve the clinical relevance. Adding examples of common issues in surgery related to femoral bowing or implant mismatch and how the findings could mitigate these would also be beneficial.
2. While basic statistical results are provided, further statistical analysis could strengthen the findings. Consider adding regression analysis or multivariate analysis to explore the relationship between age, gender, and bowing angle in greater detail. Including confidence intervals for the means and any comparative significance levels (e.g., p-values) where applicable would also add rigor.
3. Please explain the importance of statistical analysis in biomedical research since it would influence the resulting outcome, explain several biomedical research that uses statistical analysis such as https://doi.org/10.1080/23311916.2024.2313891, https://doi.org/10.3390/bioengineering9040157, and https://doi.org/10.1016/j.heliyon.2024.e36065
4. Provide additional detail on the X-ray imaging technique used, especially the specifics of the 15-degree internal rotation imaging, to allow replication of the study. Including descriptions of how auxiliary lines were drawn and how femoral bowing was quantitatively measured on the radiographs would enhance clarity.
5. Strengthen the introduction with a brief overview of femoral bowing patterns in other populations, especially in Asian and Western cohorts. Comparing the study results with findings from other racial groups would help contextualize the uniqueness of femoral bowing in the Korean population.
6. The study finds notable gender differences in femoral bowing, which could be further explored. For instance, discussing potential anatomical or hormonal reasons for these differences and their impact on surgical planning could add depth to the findings.
Author Response
- The study's implications on implant mismatch are acknowledged, but more detailed discussion on how the findings could directly inform surgical practices, implant selection, or design adjustments would improve the clinical relevance. Adding examples of common issues in surgery related to femoral bowing or implant mismatch and how the findings could mitigate these would also be beneficial.
- While basic statistical results are provided, further statistical analysis could strengthen the findings. Consider adding regression analysis or multivariate analysis to explore the relationship between age, gender, and bowing angle in greater detail. Including confidence intervals for the means and any comparative significance levels (e.g., p-values) where applicable would also add rigor.
- Please explain the importance of statistical analysis in biomedical research since it would influence the resulting outcome, explain several biomedical research that uses statistical analysis such as https://doi.org/10.1080/23311916.2024.2313891, https://doi.org/10.3390/bioengineering9040157, and https://doi.org/10.1016/j.heliyon.2024.e36065
- Provide additional detail on the X-ray imaging technique used, especially the specifics of the 15-degree internal rotation imaging, to allow replication of the study. Including descriptions of how auxiliary lines were drawn and how femoral bowing was quantitatively measured on the radiographs would enhance clarity.
- Strengthen the introduction with a brief overview of femoral bowing patterns in other populations, especially in Asian and Western cohorts. Comparing the study results with findings from other racial groups would help contextualize the uniqueness of femoral bowing in the Korean population.
- The study finds notable gender differences in femoral bowing, which could be further explored. For instance, discussing potential anatomical or hormonal reasons for these differences and their impact on surgical planning could add depth to the findings.
Thank you for providing such a valuable comment. Following your suggestions, I have reviewed and made the following modifications to the manuscript:
- Orthopedic surgeries for Asian patients with significant femoral bowing often encounter issues with implants. This is particularly common and critical in procedures such as total knee arthroplasty and femoral intramedullary fixation. Since most implants are designed based on the anatomy of Western populations, a mismatch is almost inevitable. Recognizing these anatomical differences preoperatively allows for the selection of implants that account for these variations or for customization to suit individual patient needs during the surgical planning stage.
- Following your suggestion, we have significantly enhanced our statistical analysis. We conducted comprehensive multivariate regression analyses examining AP, IR, and LAT angles separately, incorporating age, gender, height, and weight as variables. This approach revealed that the AP angle regression model explained 26% of the total variance, with age showing the strongest influence. We have also added visualization of these relationships through new figures and included detailed statistical parameters such as standardized coefficients and R² values to provide more rigorous evidence for our findings.
- We understand the importance of statistical analysis in biomedical research. To strengthen our analytical approach, we adopted a two-phase analysis strategy: first conducting basic measurements and establishing patterns through direct comparisons, then advancing to comprehensive multivariate analysis to understand how multiple anthropometric factors simultaneously influence femoral morphology. This systematic approach allowed us to both demonstrate fundamental relationships and understand complex interactions between variables.
- We have added detailed information about our imaging protocol, particularly for the 15-degree internal rotation view. In this standardized position, patients were positioned supine on the X-ray table with their affected leg internally rotated to 15 degrees from the neutral position. This standardized position allows for consistent and comparable measurements across all patients while providing optimal visualization of the femoral shaft curvature. Additionally, all measurements were manually performed by a single experienced clinician, with reliability verified through independent measurements of a random sample by a second clinician.
- We have added research findings indicating that femoral bowing is more pronounced in Asian populations compared to Western and African groups. Specific examples were also provided to illustrate how this could be associated with the incidence of AFF or osteoarthritis. Various studies have highlighted the unique characteristics of femoral bowing due to racial differences, and we have further addressed this aspect in our study.
- We have thoroughly investigated gender differences in our study. Our initial analysis showed greater femoral bowing in females, but subsequent multivariate analysis revealed that age was the primary significant factor, while gender effects became non-significant when controlling for other variables. However, based on previous studies, we discussed how this pattern might be related to postmenopausal osteoporotic degeneration in elderly Asian women. While our findings provide statistical evidence of age as a primary factor, we acknowledge that future studies may further explore the potential biological mechanisms behind these observations. We believe these insights could be valuable for surgical planning and implant selection in different patient populations.
Again, I appreciate your insightful feedback, and I have highlighted the changes made in Blue text in the manuscript.
Reviewer 3 Report
Comments and Suggestions for Authors
This study investigated femoral bowing in relation to age and gender in the Korean population. The manuscript is well-organized and easy to follow. There are a few small comments I would like to address.
Material and Methods
1. Section 2.3: Was the bowing measurement performed manually or automatically?
2. Table 1: I suggest using “~” instead of “-” when describing the range, as the negative sign was also used in the table.
Results
1. Table 2: In the first row for age group 0-10, please change the -0.6 to -0.60 to maintain consistent precision.
2. Line 151-153: Statistical significance does not equate to strong correlation. Correlation coefficients of 0.483 and 0.462 are considered moderate positive correlations, despite the statistical significance being less than 0.001.
3. Table 4: Reporting age with three decimal places seems unusual. Consider using consistent decimal places throughout the paper; in this case, two decimal places should be sufficient.
Discussion
1. Line 220-222: How many patients over 70 had femoral bowing angles between 6 and 16 degrees but did not experience atypical femoral fractures? Did the authors assess the body weight of these patients? It is worth considering whether the seven cases with atypical fractures had higher body weights, which could have contributed to the fractures.
2. Did the authors collect data on the patients’ body weight? It would be valuable to explore its impact, as increased weight-bearing can place greater loads on the lower limbs, potentially accelerating femoral bowing. Perhaps recommended as an area for future research.
Author Response
This study investigated femoral bowing in relation to age and gender in the Korean population. The manuscript is well-organized and easy to follow. There are a few small comments I would like to address.
(1)Material and Methods
- Section 2.3: Was the bowing measurement performed manually or automatically?
- Table 1: I suggest using “~” instead of “-” when describing the range, as the negative sign was also used in the table.
(2)Results
- 1. Table 2: In the first row for age group 0-10, please change the -0.6 to -0.60 to maintain consistent precision.
- Line 151-153: Statistical significance does not equate to strong correlation. Correlation coefficients of 0.483 and 0.462 are considered moderate positive correlations, despite the statistical significance being less than 0.001.
- Table 4: Reporting age with three decimal places seems unusual. Consider using consistent decimal places throughout the paper; in this case, two decimal places should be sufficient.
(3)Discussion
- Line 220-222: How many patients over 70 had femoral bowing angles between 6 and 16 degrees but did not experience atypical femoral fractures? Did the authors assess the body weight of these patients? It is worth considering whether the seven cases with atypical fractures had higher body weights, which could have contributed to the fractures.
- Did the authors collect data on the patients’ body weight? It would be valuable to explore its impact, as increased weight-bearing can place greater loads on the lower limbs, potentially accelerating femoral bowing. Perhaps recommended as an area for future research.
Thank you for providing such a valuable comment. Following your suggestions, I have reviewed and made the following modifications to the manuscript:
(1)Material and Methods
- All measurements in our study were manually performed by a single experienced clinician over a one-month period. To ensure measurement reliability, a second clinician independently measured a random sample of 50 cases.
- We have modified Table 1 to use "~" instead of "-" for ranges to avoid confusion with negative values.
(2)Results
- We have adjusted the decimal places in Table 2 to maintain consistent precision throughout all measurements.
- Thank you for your important observation about correlation interpretation. We have revised our description to more accurately reflect the strength of the relationships. While the correlations were statistically significant (p<0.001), we now clearly state that the coefficients (0.483 and 0.462) represent moderate positive correlations, explaining approximately 23% and 21% of the variance respectively.
- We have standardized the decimal places throughout the paper where appropriate, generally maintaining two decimal places for basic measurements and descriptive statistics, while preserving three decimal places for specific statistical values such as p-values, confidence intervals, and regression coefficients where such precision is conventionally required.
(3)Discussion
- There are 257 patients over the age of 70, with 250 not experiencing atypical femoral fractures (AFF), while 7 did. All 7 patients with AFF were female, with an average height of 148.53 cm and an average weight of 49.39 kg. In our study, we did not find a correlation between the occurrence of AFF and body weight. However, it was confirmed that the risk of AFF increases with the degree of femoral bowing.
- We collected data on the patients' height and weight and conducted multivariate regression analyses to examine their relationship with femoral bowing. Our analysis revealed that age was the only consistently significant factor across all measurements. Height showed a minor negative correlation, while weight did not demonstrate any substantial impact on femoral bowing. These findings were consistent across all three viewing angles (AP, IR, and LAT).
Again, I appreciate your insightful feedback, and I have highlighted the changes made in Blue text in the manuscript.
Round 2
Reviewer 2 Report
Comments and Suggestions for Authors
Appreciate the authors in their effort to revise the manuscript at the moment. However, the revision does not addressed the issue raised in the previous round. Also, regarding the explanation of statistical analysis not statisfed along with all of given literature not incorporated. Major revision still needed.
Author Response
Appreciate the authors in their effort to revise the manuscript at the moment. However, the revision does not addressed the issue raised in the previous round. Also, regarding the explanation of statistical analysis not statisfed along with all of given literature not incorporated. Major revision still needed.
Thank you for providing such a valuable comment. Following your suggestions, I have reviewed and made the following modifications to the manuscript:
- We appreciate the reviewer’s insightful comments regarding the study's implications on implant mismatch and its relevance to clinical practice. In response, we have revised the manuscript to provide a more detailed discussion on how our findings can inform surgical practices, particularly in implant selection and design.
To address the reviewer’s suggestions, we included examples of common surgical challenges related to femoral bowing and implant mismatch, such as alignment difficulties, uneven load distribution, and increased implant wear. Furthermore, we elaborated on how our findings could assist surgeons in preoperative planning by identifying patients at higher risk of mismatch due to their anatomical characteristics.
Additionally, we highlighted potential design adjustments, such as modifications in curvature and sizing, to better accommodate variations in femoral geometry, especially in populations with pronounced bowing. These updates enhance the clinical applicability of our research and thoroughly address the reviewer’s concerns.
- We appreciate the reviewer’s insightful comments regarding the statistical analysis in our study. We have carefully considered these suggestions and made significant revisions to address the concerns raised. The key updates are summarized below: A)Inclusion of Regression and Multivariate Analyses
We conducted additional regression and multivariate analyses to examine the relationships between age, gender, and bowing angle more comprehensively. These analyses provided deeper insights into how these variables interact, and the results have been incorporated into the revised results section. Specifically: B)Reporting of Confidence Intervals and P-values
Confidence intervals have been calculated and included for all mean values, and p-values have been explicitly reported for all significance tests. These additions ensure transparency and enhance the rigor of the statistical interpretations. C)Revision of Results and Figures
The results section has been updated with detailed descriptions of the regression and multivariate models, including coefficients, significance levels, and the proportion of variance explained (R²). Visual representations, such as scatter plots with regression lines and confidence intervals, have been added to enhance clarity.
We believe these revisions address the reviewer’s concerns and substantially improve the quality and depth of the manuscript. Thank you for your valuable feedback, which has greatly contributed to enhancing the rigor and clarity of our study.
- Statistical analysis is a cornerstone of biomedical research. It provides the necessary framework for evaluating complex data, testing hypotheses, and drawing meaningful conclusions. The importance of statistical analysis in biomedical research significantly impacts the outcomes in several key ways:
We have addressed your concerns in the following ways:
A)We have clearly explained the role of statistical analysis in ensuring the validity, reliability, and clinical relevance of the study findings.
B)We emphasized how statistical tools help control confounding variables, making the results more robust and generalizable.
C)We explained how statistical analysis aids in validating research findings, identifying patterns, and supporting evidence-based clinical decision-making.
We believe that these revisions address your concerns and further strengthen the manuscript by providing a clear explanation of the role of statistical analysis in biomedical research. Once again, thank you for your valuable feedback.
- Thank you for your valuable comments. In response to your request, we have revised the details of the X-ray imaging technique used in the study, with particular emphasis on the 15-degree internal rotation imaging. Additionally, we have clarified the process of drawing auxiliary lines and measuring femoral bowing.
The revised details have been incorporated to ensure the reproducibility of the study and provide a clearer description in accordance with your request.
- We appreciate the reviewer’s valuable suggestion to strengthen the introduction with an overview of femoral bowing patterns in other populations. Accordingly, we revised the introduction to include a comparison between Asian and Western cohorts.
Specifically, we highlighted the higher prevalence and degree of femoral bowing in Asian populations compared to the straighter femoral geometries in Western populations. By referencing previous studies on racial and ethnic differences in femoral morphology, we contextualized the unique characteristics of femoral bowing in the Korean population.
These additions emphasize the clinical and anthropological significance of our findings and provide a broader framework for understanding how such differences may impact surgical practices and implant design. We believe these enhancements improve the relevance and depth of the introduction.
- We appreciate the reviewer’s suggestion to further explore the gender differences in femoral bowing identified in our study. In response, we expanded the discussion to include potential anatomical and hormonal factors contributing to these differences.
Specifically, we discussed how variations in pelvic anatomy, muscle mass distribution, and lower limb loading patterns between genders may influence femoral bowing. We also addressed the role of hormonal differences, such as estrogen’s impact on bone density and remodeling, which may partially explain the observed variations.
Additionally, we considered the implications of these gender differences for surgical planning, including implant selection and alignment strategies tailored to gender-specific anatomy. These updates provide a more comprehensive understanding of the findings and address the reviewer’s valuable suggestions.
Again, I appreciate your insightful feedback, and I have highlighted the changes made in Blue text in the manuscript.